

# The relationship between price and nutritional balance for young adults in the menus of Japanese restaurants

Yu Ogasawara[1], Miyuki Asanuma[2], Masashi Kasuya[3] and Yuki Soma[4]

[1] Department of Tourism Science, Tokyo Metropolitan University, Hachioji, Tokyo, Japan
[2] Morioka Junior College, Iwate Prefectural University, Takizawa, Iwate, Japan
[3] School of Project Design, Miyagi University, Taiwa, Miyagi, Japan
[4] Faculty of Education, Hirosaki University, Hirosaki, Aomori, Japan

## ABSTRACT

**Background:** Eating habits are a contributing factor to obesity. Higher-priced menu items have better nutritional quality/balance, as the relationship between the price of food per serving and nutritional quality/balance has been reported. However, previous studies on the nutritional content of restaurant menu items did not focus on the relationship between the nutritional balance of menu items and prices. Therefore, this study aimed to investigate this relationship.

**Methods:** The nutritional balance score (NBS) was defined and calculated according to each nutritional criterion of men and women aged 18–29 years, covering more than 2,000 menu items in 26 Japanese restaurant chains. Furthermore, NBS distribution by gender and restaurant brand, and the relationship between the menu item's NBS and price were assessed.

**Results:** The results showed that the average NBS of the analyzed menu items differed between the criteria for men and women, with the menu items assessed based on men's criterion being more nutritionally balanced on average.

The compositions of the top 10 menu items differed between men and women, and most were set menus or rice bowl menus, which were offered by fast-food restaurants. The relationship between price and NBS in most fast-food and casual restaurants was expressed as a concave function. The maximum NBS based on the criteria for men and women were 64.9 and 64.1, with prices of 639.9 and 530.3 yen, respectively.

**Discussion:** NBS score increased with price to a certain level before decreasing, suggesting that the price at which NBS was the highest differed between men and women. The results of this study could contribute to the development of a methodology for healthy eating out practices, with a focus on price.

# INTRODUCTION

Eating habits are one of the factors contributing to obesity. A high-fat diet promotes weight gain (*Cao et al., 2020*), and obesity can be reduced by improving nutritional balance (*Hooper et al., 2012*). Younger people are less aware of nutritionally balanced healthy eating than older people, and cost and time are possible barriers to healthy eating in

Corresponding author
Yuki Soma, yukis@hirosaki-u.ac.jp

younger people (*Chambers et al., 2008*). Fast food is a low-cost and less time-consuming food option. Over 30% of children and adolescents in the US consumed fast food on a given day in 2015–2018 (*Fryar et al., 2020*). Fast food is often served in restaurants and is consumed away from home. In the U.S., with the exception of 2020, the year of the COVID-19 pandemic, the share of the budget for food away from home steadily increased (*United States Department of Agriculture, 2022*). In Japan, 36.2% of adults aged 20–29 years had food away from home twice or more per week, according to the National Health and Nutrition Survey in 2019 (*Ministry of Health, Labour and Welfare, 2019*). This might be because young people who leave their parents' homes for higher education or employment have more opportunities to make their own food choices, and eating out is one of them. Nutritional quality refers to the same concept as dietary quality or nutritional balance. Nutritional quality, an indicator introduced below, is used to measure the nutritional balance of food in a particular nutrient group. In this study, nutritional quality was considered similar to nutritional balance and is written in terms of nutrient balance.

Menus at fast-food restaurants include many energy-dense foods and sugary beverages, which are offered at low prices (*Beydoun et al., 2011*; *Drewnowski, 2018*; *Wilde, Llobrera & Valpiani, 2012*). However, the problem of nutritional quality in restaurant menus, which provide a food service environment, is not limited to fast food restaurants. The menus offered by large chain restaurants have been analyzed in terms of nutritional balance; family type restaurant have main entrées with higher energy, fat, and saturated fat than those in fast-food restaurants (*Wu & Sturm, 2013*). In 2018, more than 95% of the menus served by large-chain restaurants in the US and UK exceeded the recommended levels for at least one of the following items: sodium (Na), fat, saturated fat, or sugars (*Huang et al., 2022*). In Japan, the nutritional content of children's menus at chain restaurants was assessed using the Standard for the School Lunch Program, and more than half of the menu items were found to have excessive fat and salt (*Uechi, 2018*).

Even if some parts of menus in restaurants are nutritionally balanced, this does not necessarily mean that consumers actually choose those parts. While deciding what to choose among food and menu options, one of the factors that significantly influences the decision is price. Regarding the relationship between the nutritional quality of foods and their prices, low energy density foods (*e.g.*, vegetables) have higher costs per 100 g, whereas high energy density foods (*e.g.*, fats, oils, and grains) have lower costs; this difference is associated with the water content of the foods (*Drewnowski, 2018*). Although evaluating the cost of food in terms of nutrition, assessing food nutritional value per serving is the only approach directly related to the amount typically consumed, it depends on country or culture because the amount served often varies according to these factors (*Primavesi et al., 2015*). As the amount of food served is usually determined by dietary patterns, the concept of nutritional balance has shifted from individual foods to composite meals and dairy food patterns (*Drewnowski, 2018*). Therefore, dietary patterns have been analyzed in terms of nutritional balance and costs (*Carlson & Frazão, 2014*; *Drewnowski, 2018*; *Drewnowski & Eichelsdoerfer, 2009*; *Primavesi et al., 2015*). Because restaurant menus include appetizers or side menus, the serving size for one person does not always correspond to a single menu item. However, there are menus, such as set menus in fast-food restaurants, that are

designed to be served to one person. Traditional Japanese meals are also served as single set menus consisting of rice, soup, a main dish, and a side dish (*Gabriel, Ninomiya & Uneyama, 2018*). Thus, menus that correspond to meals serving one person are often seen in Japanese restaurants. Because side menus, including appetizers, which can be combined with other menu items, are offered in smaller quantities (*Wu & Sturm, 2013*), we can assume that the prices of these menu items are lower. In addition, a lower-priced menu may be less nutritionally balanced because of smaller portions, fewer ingredients, and cheaper food sources. Conversely, a higher-priced menu might be nutritionally well balanced as it may include higher-priced foods or a set menu for one person comprising various dishes. This leads to the hypothesis that a higher-priced menu item has better nutritional balance. With regard to food items, *Primavesi et al. (2015)* calculated scores indicating nutritional balance for several foods and showed a relationship between nutritional balance and the price of the foods per 100 g and recommended the serving size, showing that a higher-price food item has a better nutritional balance per recommended serving size. However, previous studies on the nutritional content of restaurant food have not focused on the relationship between the nutritional balance of menu items and prices. In addition, it would be difficult to collect nutrition data with sufficient items from various restaurant menus to use the existing measures.

In this study, we defined a new measure of nutritional balance, namely, nutritional balance score, that could be applied to the nutritional data of meals with any number of nutritional items, and calculated this score for each nutritional criterion for men and women aged 18–29 years, covering more than 2,000 menus in 26 Japanese restaurant chains. We then determined whether the relationship between nutritional balance and food prices, in which nutritional balance improves as price increases, also exists for meals served in restaurants. Our findings could contribute to the development of price-focused healthy eating methodologies and provide insights into the pricing of healthy meals in restaurants.

## MATERIALS AND METHODS

### Data collection

Data were collected from 478 chain restaurants listed in an overview of the publicly listed food service companies in 2020 (*Food Business Research Institute, Limited, 2019a*) and 50 Japanese restaurant chains (*Food Business Research Institute, Limited, 2019b*), excluding 27 brands that only opened stores outside Japan, 85 brands for which the number of stores for each brand was unknown because the brand name was listed as "Other" or the number of stores was listed as the sum of multiple brands, and 46 duplicate brands, resulting in a total of 370 brands. Nutritional data were collected from official restaurant websites between October 2020 and January 2021. The results showed that 98 restaurant chains published nutritional data on their websites. In this study, protein (g), fat (g), carbohydrate (g), and salt (g) levels were used to assess nutritional balance. Energy was not employed to calculate the nutritional balance because it depended on the values of protein (g), fat (g), and carbohydrates (g). Of the 98 chain restaurants with published nutritional data on their websites, 26 had information on these nutritional items and prices for their menu items.

In this study, 26 chain restaurants (fast-food restaurants: 13, casual restaurants: seven, café-type restaurants: five, pub-type restaurants: 1) were analyzed. Chain restaurants had 2,391 menu items which did not include beverages.

## Nutritional criteria

The definition of "nutritionally balanced menu" varied depending on the source and context, similar to that of "healthy food" (*Primavesi et al., 2015*). This study employs values for a tentative dietary goal (DG) for preventing life-style related diseases in people aged 18 to 29 years from the Dietary Reference Intakes for Japanese 2020 (*Ministry of Health Labour and Welfare, 2019*) as the nutritional criteria for protein, fat, and carbohydrate. DG has upper and lower bounds, that is, a range. Physical activity Level I was used to identify the DG values for men and women. As DG is presented as daily intake, it is necessary to determine the percentage of nutrients consumed in a day from meals consumed at restaurants. In the absence of a standard method for determining this percentage, it was set to 40% in this study. In the Dietary Reference Intakes for Japanese 2020 (*Ministry of Health Labour and Welfare, 2019*), the DG values of fat and carbohydrates are listed in kcal; therefore, the standards listed in this document (9 kcal/g fat and 4 kcal/g carbohydrates) were converted to grams. The DG of salt for people aged 18–29 is provided as a range with only the upper bound in the Dietary Reference Intakes for Japanese 2020. In this study, the lower bound of the salt criterion was based on the estimated average requirement (EAR) listed in the Dietary Reference Intakes for Japanese 2020. The criteria for the nutritional items are listed in Table 1.

## Nutritional measurement

Several measures to evaluate the nutritional quality of foods have been suggested, including the nutrient-rich (NR) subscore, mean adequacy ratio (MAR), mean excess ratio (MER), nutritional quality (NQ), and Rudd Modified Nutrient Profile Index (MNPI), which is the proposed method to determine whether a diet is healthy. There are four types of NR subscores, calculated from 6, 9, 11, and 15 nutrient variables (*Drewnowski, Maillot & Darmon, 2009*). NQ is calculated from MAR and MER. The MAR is calculated as the mean percentage of the daily recommended intake of some beneficial nutrients for each diet or one food. The number of nutrients to calculate a score for MAR is greater than the number of nutrients to calculate a score for NR. *Primavesi et al. (2015)* used energy density and 28 nutrients and *Vieux et al. (2013)* used 20 nutrients to calculate MAR. MER is an indicator of bad nutritional quality and is calculated from three nutrients; fats, free sugars, and sodium (*Primavesi et al., 2015*; *Vieux et al., 2013*). MNPI is based on the UK-Ofcom nutrient profiling model using a point system of health and unhealthy food components (*Lesser et al., 2017*).

However, sufficient data could not be collected from Japanese restaurant menus to use the existing measures, and some of the existing nutritional scores are based on non-Japanese guidelines. Specifically, price, protein, fat, carbohydrate, and salt were listed on 66.0%, 39.9%, 39.9%, 39.0%, and 53.3% of the meals, respectively, in the 98 chain

**Table 1 The criteria for the nutritional items.**

| Item (g) | Men | Women |
|---|---|---|
| Protein | [30.0, 46.0] | [22.8, 35.2] |
| Fat | [20.4, 30.7] | [15.1, 22.7] |
| Carbohydrate | [115.0, 138.0] | [85.0, 110.5] |
| Salt | [0.6, 3.0] | [0.6, 2.6] |

**Note:**
Upper and lower bounds for protein, fat, carbohydrate, and an upper bound for salt were set by the tentative dietary goal for preventing life-style related diseases from the Dietary Reference Intakes for Japanese 2020. A lower bound for salt was set by the estimated average requirement in the Dietary Reference Intakes for Japanese 2020. The values of the criteria are calculated as 40% of the nutrients consumed in a day for the 18 to 29 years age group, using Physical activity Level I. Numerical conversions to grams were made using 9 kcal/g fat and 4 kcal/g carbohydrate.

restaurant menus. Other than these nutrient items, fibre was the most frequently listed nutrient on the menus; however, it was listed in only 7% of the total meals.

Thus, this study provides a general nutritional measurement called the nutritional balance score (NBS) to evaluate the nutritional balance using the upper and lower limits of the range of nutritional criteria.

Let $m$ be the number of nutritional criteria for evaluation. The nutritional criterion $Y$ for a nutrient $p \in \{1, \cdots, m\}$ is provided as the range $Y_p = \left[\underline{Y}_p, \bar{Y}_p\right]$. $\underline{Y}_p$ and $\bar{Y}_p$ are the lower and upper bounds, respectively, for the range $Y_p$. Let $n$ be the number of observed data points, that is menu items. The nutrition of the data $i \in \{1, \cdots, n\}$ is denoted as $x_{i\cdot} = (x_{i1}, x_{i2}, \cdots, x_{im})$. It is assumed that the values of $x_{i\cdot}$ are standardized, and the nutritional criteria are adjusted to the standardized scales, i.e., $\underline{Y}_p = \left(\underline{Y}'_p - \overline{x'}_{\cdot p}\right)/\widehat{x'}_{\cdot p}$ and $\bar{Y}_p = \left(\underline{Y}'_p - \overline{x'}_{\cdot p}\right)/\widehat{x'}_{\cdot p}$, where $\overline{x'}_{\cdot p}$ and $\widehat{x'}_{\cdot p}$ are mean and standard deviation of original (unstandardized) observed data for nutrition $p \in \{1, \cdots, m\}$. The sum of the unweighted difference between nutritional criterion $Y$ and data $x_{i\cdot}$ is defined as follows:

$$s'_Y(x_{i\cdot}) = \sum_{p=1}^{m} d(x_{ip}, Y_p)$$

where

$$d(x_{ip}, Y_p) = \begin{cases} x_{ip} - \bar{Y}_p, & \bar{Y}_p \leq x_{ip} \\ 0, & \underline{Y}_p < x_{ip} < \bar{Y}_p \\ \underline{Y}_p - x_{ip}, & x_{ip} \leq \underline{Y}_p \end{cases}.$$

We modified $s'_Y(x_{i\cdot})$ to be more interpretable, and the changed measurement was called the Nutritional Balance Score (NBS) in this study. The NBS for data $x_{i\cdot}$ based on criterion $Y$ is defined as $\hat{s}_Y(x_{i\cdot}) = -s'_Y(x_{i\cdot}) + s'_Y(0)$ where 0 indicates a $p$-dimensional zero vector. Furthermore, adjusted NBS is defined as $s_Y(x_{i\cdot}) = 100 \times (-s'_Y(x_{i\cdot}) + s'_Y(0))/s'_Y(0)$. The suggested measurements, NBS and adjusted NBS, have the following features: 1) a higher (adjusted) NBS means a more nutritionally balanced menu under criterion $Y$, 2) $-\infty < \hat{s}_Y(x_{i\cdot}) \leq s'_Y(0)$ and $-\infty < s_Y(x_{i\cdot}) \leq 100$, 3) $\hat{s}_Y(x_{i\cdot}) = 0$ and $s_Y(x_{i\cdot}) = 0$ indicates the case where nothing is eaten, 4) if $\hat{s}_Y(x_{i\cdot}) \leq \hat{s}'_Y(x_{i\cdot})$ for a given menu $x_{i\cdot}$ under two different nutritional criteria $Y$ and $Y'$, then the evaluation of the menu $x_{i\cdot}$ is higher when
based on criterion $Y$ than that based on criterion $Y'$. From 3), a negative (adjusted) NBS indicated that the menu item was worse than eating nothing. The adjusted NBS is modified from the NBS so that the upper limit is 100, which means that we cannot compare the adjusted NBSs calculated by different criteria, whereas we can compare NBSs under that situation, as shown in feature 4).

To show the relationship between NBS and price, in addition to scatter plots, we use regression lines obtained from the generalized additive model with Gaussian distribution. The calculation was performed using the mgcv package ver.1.8 on R ver.4.1.

## RESULTS

### NBSs for brands based on criteria of men and women

Table 2 shows the basic statistics of price and the nutrition items for the collected original (unstandardized) menu data. In addition, the adjusted criteria for the nutrient items calculated by mean and standard deviation of original menu data are shown in the table. Given that nutritional data for analyzing are standardized, and mean and standard deviation for each item are 0 and 1, respectively, means of protein and carbohydrate are not above the minimum of the adjusted criteria for men and women. This may be due to the inclusion of appetizers and other small side dishes on the menu data. With regard to fat, the mean fell within the range for men, but not for women. For both men and women, the range of the criterion of salt was lesser than zero, which indicates that the menu items including appetizers tended to contain more salt than defined in the criterion.

Let $M$ and $F$ be the criteria for men and women, respectively. The NBSs of menus for the $M$ and $F$ criteria are shown in Fig. 1. Means and standard deviations of NBSs for $M$ and $F$ are 2.45 (SD, 1.53) and 1.61 (SD, 1.62), respectively. NBSs for zero vector for M and F, which indicate the upper limits of the NBSs and scales while calculating adjusted NBSs, are $s'_M(0) = 5.40$ and $s'_F(0) = 4.09$, respectively. These results suggest that the menus meet more of the nutritional balance criteria for men than those for women, even though the same menus are provided to both sexes. In addition, many menu items have an NBS < 0, which is worse than eating nothing. Figure 2 shows the NBSs of menus for each brand for M and F. Brands 1–13, 14–20, 21–25, and 26 correspond to fast food restaurants, casual restaurants, café-type restaurants, and pub-type restaurants, respectively. The NBSs and their interquartile ranges of café- and pub-type restaurants are relatively lower and narrower than those of other restaurants, respectively. Brand 5 had the same characteristics. This may be because Brand 5 is a sushi restaurant that serves one plate as a menu item at a time (*i.e.*, conveyor belt sushi). The top 10 items of the adjusted NBS based on the criteria for men and women are listed in Table 3. The compositions of the top 10 menus differed between men and women, and most menus were set menus or rice bowl menus offered by fast-food restaurants.

### NBSs and price

The relationship between NBS and price is shown as a scatter plot and a regression line (Fig. 3). We have used the adjusted NBSs in the figure. The maximum NBS values based on the criteria for men and women were 64.9 and 64.1, respectively, and the prices to achieve

**Table 2 Basic statistics for nutrition data items.**

| Item | Mean | S.D. | Adjusted criteria | |
|---|---|---|---|---|
| | | | Men | Women |
| Price (yen) | 664.43 | 541.2 | – | – |
| Protein (g) | 21.21 | 15.91 | [0.55, 1.56] | [0.10, 0.88] |
| Fat (g) | 25.57 | 20.16 | [−0.25, 0.25] | [−0.52, −0.14] |
| Carbohydrate (g) | 64.73 | 50.51 | [1.00, 1.45] | [0.40, 0.91] |
| Salt (g) | 3.35 | 2.71 | [−1.01, −0.13] | [−1.01, −0.28] |

Note:
Means and S.D.s were calculated from the 2,391 menu items analyzed. The adjusted criteria are the values of the criteria in Table 1 standardized by means and S.D.s.

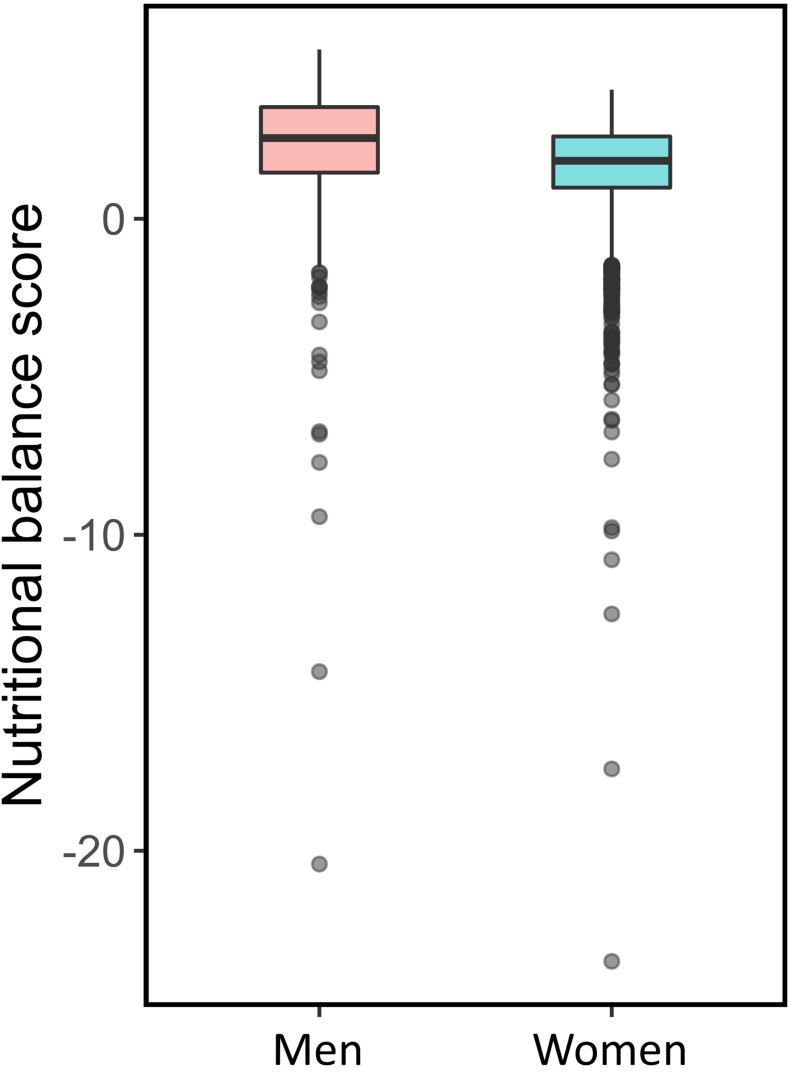

**Figure 1 Nutritional balance scores for the criteria for men and women.**

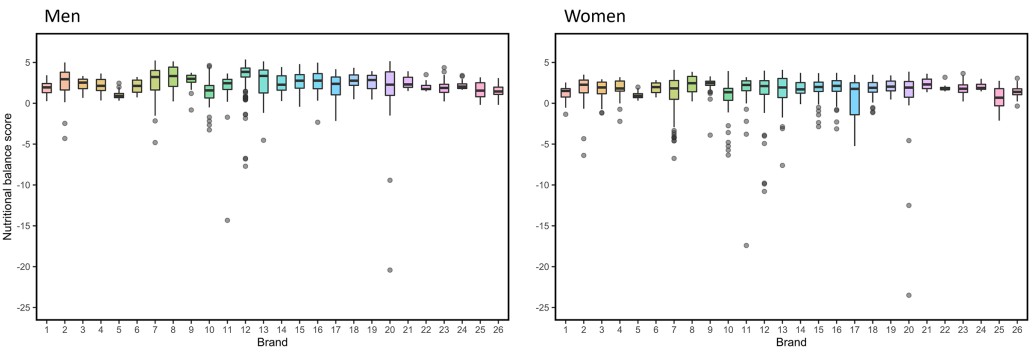

**Figure 2 Nutritional balance scores of menus by brand number for the criteria for men and women.**
Brand 1–13: fast food restaurants, Brand 14–20: casual restaurants, Brand 21–25: café-type restaurants, Brand 26: pub-type restaurants.

**Table 3 Top 10 menus for the criteria for men and women.**

| Rank | Men Brand name | Menu | Adjust NBS | Women Brand name | Menu | Adjust NBS |
|---|---|---|---|---|---|---|
| 1 | Brand 12 | Fried chicken bento extra rice | 99.32 | Brand 7 | Onion salmon bowl (regular) | 100.00 |
| 2 | Brand 12 | Fried chicken bento extra rice with boiled barley | 99.32 | Brand 13 | Eel bowl | 100.00 |
| 3 | Brand 12 | Mackerel simmered in miso bento extra rice | 98.63 | Brand 12 | Natto morning set menu extra rice with boiled barley | 97.80 |
| 4 | Brand 12 | Mackerel simmered in miso bento extra rice with boiled barley | 98.63 | Brand 12 | Beef bowl (regular) | 97.17 |
| 5 | Brand 12 | Salt-grilled mackerel and fried chicken bento extra rice with boiled barley | 97.27 | Brand 10 | Margherita with fresh tomato small size | 96.50 |
| 6 | Brand 12 | Salt-grilled mackerel and fried chicken bento extra rice | 97.23 | Brand 7 | Chopped minced chicken bowl (large) | 96.50 |
| 7 | Brand 7 | Tuna-Yukke bowl (large) | 97.11 | Brand 7 | Chopped minced chicken bowl (regular) | 96.35 |
| 8 | Brand 7 | Beef bowl (large) | 96.01 | Brand 10 | Margherita with 5 kinds of cheese small size | 96.10 |
| 9 | Brand 12 | Ginger grilled pork bento extra rice with boiled barley | 95.87 | Brand 12 | Salt-grilled salmon morning set menu extra rice with boiled barley | 95.49 |
| 10 | Brand 20 | Original lunch set menu | 95.49 | Brand 13 | Salt-grilled mackerel and natto set menu | 95.33 |

them were 639.9 and 530.3 yen, respectively. It is natural for graphs to show multiple peaks because they include different types of brands.

Three examples from the scatter plot by brand are shown in Fig. 4. In the figure, only scores calculated from the criterion for men are included because the figures for the criteria for men and women were similar (Fig. 3). The brands in the examples are Brand 7, Brand 16, and Brand 22, which correspond to fast-food, casual, and café-type restaurants, respectively. All graphs by brand for men's and women's criteria are provided as Supplemental Materials. Brand 26 was not included in the Supplemental Materials because

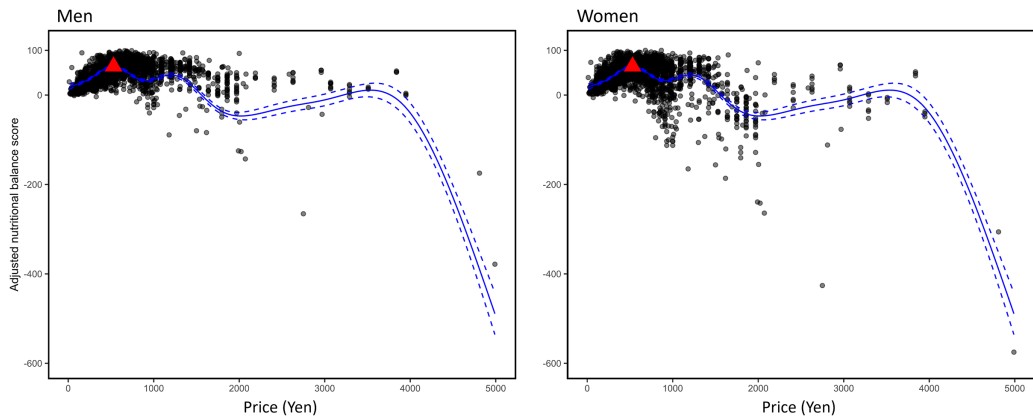

**Figure 3 Adjusted NBSs and prices calculated from the criteria for men and women.** The dashed lines represent the upper and lower bounds of the 95% confidence intervals. The maximum value for each graph is represented by a triangular point.

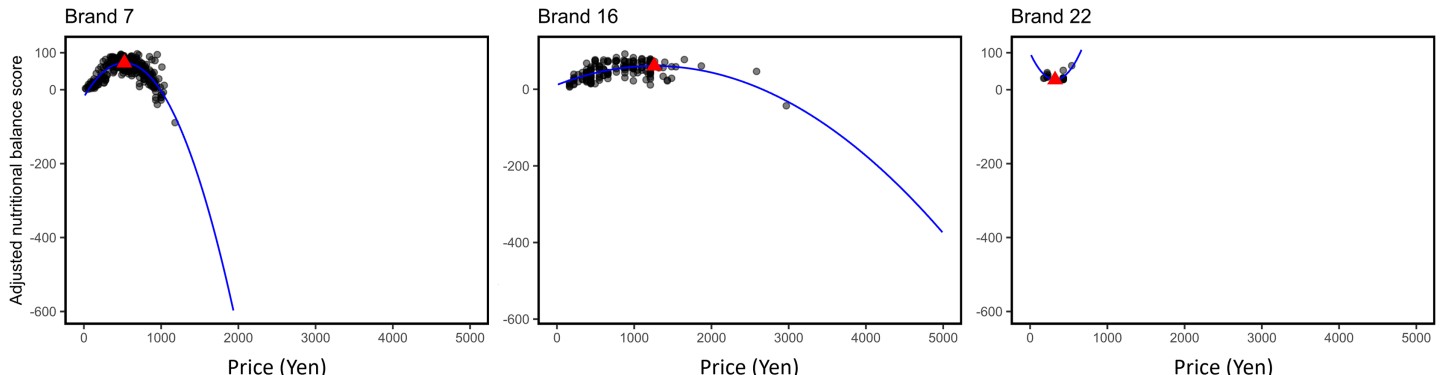

**Figure 4 Example scatter plots of brands for the criteria for men.** The maximum and minimum value points for each graph are represented by triangular points.

all the menu items were provided at the same price. Almost all relationships between adjusted NBSs and price for brands were concave, and the objective menus in each brand were all menus, excluding beverages. Thus, the regression lines were calculated using quadratic functions, and the confidence intervals are not shown. The maximum points are denoted by the triangular points.

For most brands of fast-food and casual restaurants, adjusted NBSs had concave relationships with price for both men and women; that is, adjusted NBSs increased as prices increased, but the slopes of the increase became smaller. However, such features were not observed in Brands 1, 5, 19, 21, 22, and 24. The regression lines for these brands were convex.

## DISCUSSION

This study examined the relationship between nutritional balance and price using data on the protein, fat, and carbohydrate contents of menus published by Japanese restaurant chains. The results showed a concave curve association between nutritional balance and prices.

As the menus in this study included appetizers and side dishes, the mean of each nutrient may have been lower than the range of the adjusted criteria, which considers a single meal. This may explain why the means for protein and carbohydrates were lower than the lower limits of the adjusted criteria for men and women. Considering this, it is possible that the main entrées in the menus were higher in fat and salt than the criteria. The tendency for fat and salt to exceed the recommended levels on children's menus in Japanese restaurants might be similar to the overall trend in restaurant menus in Japan (*Uechi, 2018*). This trend, and the fact that the criteria ranges of nutrient items for men were higher than those for women, might have resulted in higher NBSs for men than those for women. This also affected the differences in the top 10 menu items between men and women.

As for NBSs by brand, there was no clear difference in the distribution of NBSs between fast-food restaurants and casual restaurants, but the menus offered by café restaurants tended to have lower NBSs than those offered by other types of restaurants. This may be because most meal menus offered by café-type restaurants were single items, such as sandwiches or deserts, rather than set menus.

A concave trend was observed between price and NBS: the higher the price, the higher the NBS; however, the slope decreased, and after a certain price, the NBS started to decrease. This might be because the value of each nutritional item of a menu increases as the quantity of food comprising the menu increases with price, and the NBS decreases when the price and quantity exceed a certain level. The correlation coefficient between price and calories for menus was 0.68, which was significant at the 1% level. As the ranges of criteria for men are higher than those for women, the peak of the NBS for women is first reached when the price increases. The prices at which the peak is realized were 639.9 and 530.3 yen for the citeria for men and women, respectively. These prices, including those of appetizers and side dishes, were lower than the mean price (664.43 yen) in this study. Using the average cost of eating out per household with persons under the age of 29, the average number of household members under the age of 29, and the frequency of eating out in households with two or more household members, where the head of the household is 29 years old or younger in 2020, 986.39 yen can be estimated as the eating-out expense per person under 29 years old per occasion (*Statistics Bureau of Japan, 2021a*, *2021b*). This also suggests that the prices at which these peaks are realized are relatively low. It is possible that young people eat too much or pay more for value-added services rather than their diet while eating in restaurants. Furthermore, this might be partly due to the limited number of nutrients analyzed in this study. The NBS did not include items for vitamins, minerals, or dietary fibers. If these items were included, the menu would need to include higher-priced foods such as vegetables and fruits to add the necessary nutrients, which could increase the price of reaching the peak NBS score. Thus, including a higher number of nutrients in the NBS may increase the cost of reaching the peak score.

A concave relationship between price and NBS was observed for many brands. The menus of café-type restaurants, which did not show concave functions in some cases, have a large number of light meals (*e.g.*, sandwiches) and dessert menus with a low quantity of each nutrient, which leads to a linear relationship in which the quantity and

NBS increase, which might explain why the regression lines became convex functions. The other brands with convex functions were restaurants specializing in a limited menu of Japanese noodles with soup (Ramen) or fried rice, which could have been the reason why the relationship between price and NBS for the brands did not turn out to be concave functions. When all meals are offered at the same price, one would expect their NBSs to have similar values. Indeed, the interquartile range of the NBSs for Brand 26, where all meals are served at the same price, is relatively narrow. For brands where the regression line is a concave function, the gradient of the function (*e.g.*, the coefficient of the quadratic term) varies from brand to brand, leading to a change in the price at which the peak of the NBS is reached. This gradient indicates the influence of price on nutritional balance and reflects the composition of the menu offered by the brand. This implies that not only the mean number of nutrients and nutritional balance in the menu but also the price at which the most nutritionally balanced menu is achieved and the function indicating the relationship between nutritional balance and price are characteristics of the brand. For example, the gradients of the functions of fast-food restaurants tend to be larger than those of casual restaurants for brands with concave functions, which may be reflected in the table turnover. In restaurants with low turnover and high cost per customer, menu price has little effect on the nutritional balance of the menu, whereas in restaurants with relatively high turnover and low cost per customer, such as fast-food restaurants, menu prices may have a stronger effect on menu content (nutritional balance). Furthermore, this study shows that, while considering the relationship between price and nutritional balance in restaurant menus, not only a linear relationship, in which increasing prices increase nutritional balance, but also a more complex, nonlinear relationship should be considered.

Considering that men and women are served the same meal in a chain restaurant even though the nutritional criteria for men and women are different, the optimal price obtained by the women's nutritional criterion for a given brand is lower than the optimal price obtained by the men's nutritional criterion. Thus, for a given menu brand, women may be able to obtain a more nutritionally balanced meal than men for less money. This sex difference in optimal price varies across brands, and this difference is driven by differences in the concave function for price and NBS across brands. In this study, the mean differences in optimal price between men and women at fast-food restaurants and casual restaurants, where the fitted functions are concave functions for both men and women, were 162.7 and 234.9 yen, respectively. The difference in the optimal price for men and women when eating at a fast-food restaurant with a large concave slope was smaller than the price difference between men and women when eating at a casual restaurant with a small concave slope. Therefore, using the nutritional balance calculated from the nutritional criteria for men and women used in this study, fast-food restaurants may offer menus with smaller differences in optimal prices for men and women than casual restaurants in Japan. However, it should be noted that this difference in restaurant type was observed in a limited sample of 11 fast-food and six casual restaurants.

A limitation of this study was the small number of nutritional items. As mentioned above, this may have led to an overestimation of the NBSs in this study. While the items listed in this study are required to be labeled according to the Food Labeling Act,

information regarding items related to fiber, vitamins, and minerals was not provided in the websites of almost chain-restaurants investigated in this study. Even for fiber, which had the least amount of missing data among fiber and vitamins and minerals, only eight of the 98 restaurants listed fiber values on their menus. In addition, NBSs were calculated for single menu items; however, in actual restaurants, it is possible to select multiple menu items in combination. This would allow, for example, a single salad to be added as a side dish to a menu that does not reach the peak NBS; thus, even though the price is higher, the NBS peak is reached by a higher NBS. Such combinations make it necessary to consider more complex combinatorial optimization problems.

## CONCLUSIONS

The average nutritional balance score of the analyzed menus differed between men and women, with men's menus being more nutritionally balanced on an average. Most menus with high nutritional balance scores were traditional Japanese set menus or Japanese fast-food bowl style menus. The relationship between the NBS and price showed that the score increased up to a certain level as the price increased and then decreased, suggesting that the price at which the NBS was the highest differed between men and women. In the dataset used in this study, the price at which the highest nutritional balance score was achieved was 1.21 times higher for men than that for women. The results of this study could contribute to the development of a methodology for healthy eating out, with a focus on price.

## ACKNOWLEDGEMENTS

We would like to thank Editage for English language editing.

### Funding

This work was supported by the Strategic Research Fund (Iwate Prefectural University) under grant [number 20ZK-06, 2020–2021]. The funders had no role in study design, data collection and analysis, decision to publish, or preparation of the manuscript.

### Grant Disclosures

The following grant information was disclosed by the authors:
Strategic Research Fund (Iwate Prefectural University): 20ZK-06, 2020–2021.

### Competing Interests

The authors declare that they have no competing interests.

### Author Contributions

- Yu Ogasawara conceived and designed the experiments, performed the experiments, analyzed the data, prepared figures and/or tables, authored or reviewed drafts of the article, and approved the final draft.

- Miyuki Asanuma conceived and designed the experiments, authored or reviewed drafts of the article, and approved the final draft.
- Masashi Kasuya conceived and designed the experiments, authored or reviewed drafts of the article, and approved the final draft.
- Yuki Soma conceived and designed the experiments, performed the experiments, prepared figures and/or tables, authored or reviewed drafts of the article, and approved the final draft.

## Data Availability

The raw measurements are available in the Supplemental File.

## Supplemental Information

Supplemental information for this article can be found online at http://dx.doi.org/10.7717/peerj.18091#supplemental-information.

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
