# Peer review of "The relationship between price and nutritional balance for young adults in the menus of Japanese restaurants"

_PeerJ, doi:10.7717/peerj.18091_

## Round 0.1 · original submission · Major Revisions

As addressed by the referees, and I share their opinion, this article provides novel and relevant infrormation on a topic of public intenrest. At the same time, several concepts and methodological aspects should be better explained. Plese read the referees' comments and respond to them point by point, even if you decide not to include some of their suggestions.

Besides, I would like to apologize for the long time handling this manuscript, but it was really hard to find available referees.

Reviewer 1 ·

Basic reporting

1. One of the key concepts in the manuscript is 'nutritional balance.' However, it is not clearly defined in either the abstract or the background. How is nutritional balance typically defined, measured, and used in the literature, and why is investigating this concept important? I had to read the background in the abstract several times to understand the novelty of the study, so please revise to clarify the difference between nutritional quality and balance, and avoid using these terms interchangeably.

2. Lines 140-143: If you have the space, please briefly introduce each method. It is not clear what they entail and what is meant by 'sufficient data'.

3. Lines 232-234: I do not follow the logic here. Just because menu items in the US have higher salt and fat content, why would menu items in Japan necessarily follow the same trend? Is this observed in your data?

Other minor typos:
Line 78: "country" not "county"?
Line 124: DG as an abbreviation should be listed after dietary goals (DG).
Line 143: extra 0 in the reference
Lines 193-194: brands 1-13?

Experimental design

1. Lines 108-109: How were these chains and restaurants selected? I would expect the number of publicly listed food services to exceed 370 chains. There is no need for adjustment if you included all from the list, but if certain chains were chosen, please describe and justify the sampling method.

2. Lines 115-116: Of these 98, 26 provided nutrition information on all nutrients necessary for calculating nutritional balance and also included price information. How many of them actually have 1) nutritional values and 2) price information?

3. Lines 129-130: The authors assume that 40% of nutritional intake comes from restaurant food, which seems quite high. What is the basis for this assumption? Have you tested other percentages to assess the robustness of this assumption? In my cross-sectional sample in the UK, only 5% (median percentage) of the food was consumed out-of-home. I understand that this may differ in Japan, but further clarification would be helpful.

4. Lines 135-136: Why create a lower bound for salt? Is it necessary to have a lower bound, as it is not clear whether lower salt equals worse nutritional balance? Considering the formulas used to calculate the nutritional score, it seems possible to modify the formula for salt—if the value is lower than the upper bound, then d(Xip, Yp) =0.

5. Lines 204-205: this should be in the method section.

Validity of the findings

1. Lines 190-191: The interpretation of the results suggests that menus are better for men than for women, even though they are the same set of menus for both genders. The difference arises from the nutritional criteria used for men and women (e.g., the dietary intake upper limit is higher for men). Please review and ensure that the language throughout correctly interprets the results.

2. Figure 2: Please add a caption or group the brands directly on the graph (i.e., adding different type of restaurants such as fast food restaurants), rather than just presenting numeric values on the x-axis. The graph should also be standalone. I would also suggest grouping the results by type of restaurant rather than by brand for easier interpretation (since we don't know the exact brand).

3. Table 3: It is hard for me to believe that the fried chicken bento actually has the highest NBS score! (I would imagine fried chicken to be high in salt and saturated fat).

4. Lines 228-229: Why include appetisers and sides? These are not normally consumed as a meal. What do you consider constitutes a meal?

5. Lines 305-306 need rephrasing; I am still not quite clear on how the study findings can contribute to developing strategies for eating out. This requires a bit more thinking and should also be addressed in the discussion section.

Additional comments

The authors have done a good job in providing context, reporting their methods, and presenting their results. However, there are areas that could be improved to enhance clarity for readers. Some justifications are necessary to understand the rationale behind certain analytical choices.

·

Basic reporting

The article addresses an interesting and unaddressed topic. It is very commendable for the field work involved and for working with so much information.
However, some changes would make the article more understandable to the reader.
I would remove the first paragraph, concerning what obesity is. It would directly emphasize the second paragraph: eating habits....it is not very clear from the objectives whether all types of restaurant chains will be analyzed, or exclusively fast-food restaurants or those offering family menus.

Experimental design

The research question is good and the experimental design is correct, although it could be explained a little better.

The authors could make a scheme or flow chart clearly showing the criteria for choosing the restaurants and which ones were finally included in the study (of the initial 370, 26 were left). It is not clear if Branch 26 was finally included in the study or not, because in the results section it is stated that all the menus had the same price. If so, it would have to be taken out of all the figures.

Validity of the findings

Tables should be self-explanatory. In this sense, the title of Table 1 (The criteria for the nutritional item) does not explain the variables it reports, nor what criteria it is about. Although it is explained in the text (estimated average macronutrient requirement according to the Dietary Reference Intake for Japan 2020?).
The same can be said for the figures

Additional comments

The menus are not men's and women's menus. I understand that the restaurant does not offer different menus for men and women. It offers the same menu. It is another matter whether the menu meets the nutritional requirements better for men than for women (or vice versa). In this sense the authors need to revise the wording throughout the text, tables and figures.

I suggest further review to make sure you understand the questions raised, but I congratulate you on your work which has been a pleasure to evaluate.

Reviewer 3 ·

Basic reporting

The text presents clear and unambiguous language throughout.
Problem well contextualized in the introduction, presenting relevant literature on the topic, however the last paragraph, which could understand the objective of the research, presents part of the method: “In this study, we defined a score that indicates nutritional balance, namely the nutritional balance score, and calculated this score for each nutritional criterion of men and women aged 18.29 years, covering more than 2,000 menus in 26 Japanese restaurant chains. Furthermore, the distribution of nutritional balance scores by gender and restaurant is shown, and the relationship between the menu’s nutritional balance score and price is analyzed. This study identifies the price that achieves the best nutritional balance in each restaurant chain.s menu”.
 The structure complies with standards.
 The figures are relevant and well described.
 The raw data was provided.

Experimental design

Original primary research within scope of the journal.
 Research fills a gap in the literature, as it relates nutritional quality and the price of menu items offered in different restaurants.
 The investigation met technical and ethical research standards.
 Methods presents sufficient information to allow replication of the research.

Validity of the findings

The underlying data has been provided and clarifies how the results were presented..
 The conclusions are well formulated and appropriate to the research objective. Shows the relationship between nutritional quality and price of Japanese restaurant menus. The research points to interesting conclusions: 1) Most menus with high nutritional balance scores were traditional Japanese menus or fast-food style Japanese menus; 2) men's menus, on average, were more nutritionally balanced than women's menus; 3) There is a positive relationship between nutritional balance and price of menu components and 4) the price at which the highest nutritional balance score was achieved was 1.21 times higher for men than for women. This and other findings from this research can contribute to the development of strategies for providing healthy food outside the home.

Additional comments

Obesity is recognized as a complex chronic disease and although the causes and interventions needed to mitigate the problem are well understood, there is still a need for investment in interventions that result in the minimization of this epidemic. At the World Health Assembly in 2022, Member States adopted the WHO Acceleration Plan to contain obesity by 2030. The strategies advocated by the WHO to contain obesity rates include several actions that support healthy practices such as: o breastfeeding support; regulations on advertising food and beverages to children; school food and nutrition policies, including initiatives to regulate the sale of products rich in fat, sugar and salt near schools; tax and price policies to promote healthy diets; nutritional labeling policies; education and awareness campaigns for healthy diets and exercise; promoting physical activity in schools and integrating obesity prevention and management services into primary health care.
This research aims to investigate the relationship between the nutritional balance of menu items offered in Japanese restaurants and the prices of these items, given that eating outside the home has grown in several countries and eating habits are a risk factor for obesity.

---

## Round 0.2 · accepted · Accept

I have read the rebuttal letter provided aby the authors, the modified version of the manuscript and the new assessment performed by one of the reviewers (the other ones were not available).

Based upon this, I consider the authors have properly addressed all the questions provided by the reviewers as well as the modifications which were suggested. So I consider the mnauscript is acceptable for publication.

·

Basic reporting

NO COMMENT
The authors have implemented all the changes recommended in the previous revisions. I have no further comments to add.

Experimental design

NO COMMENT
The authors have implemented all the changes recommended in the previous revisions. I have no further comments to add.

Validity of the findings

NO COMMENT
The authors have implemented all the changes recommended in the previous revisions. I have no further comments to add.

Additional comments

Dear Authors,
I would like to congratulate you on your efforts. The article meets the standards required for publication.